# Functional Interactomes of Genes Showing Association with Type-2 Diabetes and Its Intermediate Phenotypic Traits Point towards Adipo-Centric Mechanisms in Its Pathophysiology

**DOI:** 10.3390/biom10040601

**Published:** 2020-04-13

**Authors:** Aditya Saxena, Nitin Wahi, Anshul Kumar, Sandeep Kumar Mathur

**Affiliations:** 1Department of Biotechnology, Institute of Applied Sciences and Humanities, GLA University, Mathura 281 406, India; aditya.235@gmail.com; 2Department of Bioinformatics, Pathfinder Research and Training Foundation, Gr. Noida 201 308, India; 3Department of Endocrinology, Sawai Man Singh Medical College and Hospital, Jaipur 302 004, India; anshul.singh2910@gmail.com (A.K.); drsandeepmathur@rediffmail.com (S.K.M.)

**Keywords:** type 2 diabetes, HOMA-β, HOMA-IR, genenetworks

## Abstract

The pathogenic mechanisms causing type 2 diabetes (T2D) are still poorly understood; a greater awareness of its causation can lead to the development of newer and better antidiabetic drugs. In this study, we used a network-based approach to assess the cellular processes associated with protein–protein interaction subnetworks of glycemic traits—HOMA-β and HOMA-IR. Their subnetworks were further analyzed in terms of their overlap with the differentially expressed genes (DEGs) in pancreatic, muscle, and adipose tissue in diabetics. We found several DEGs in these tissues showing an overlap with the HOMA-β subnetwork, suggesting a role of these tissues in β-cell failure. Many genes in the HOMA-IR subnetwork too showed an overlap with the HOMA-β subnetwork. For understanding the functional theme of these subnetworks, a pathway-to-pathway complementary network analysis was done, which identified various adipose biology-related pathways, containing genes involved in both insulin secretion and action. In conclusion, network analysis of genes showing an association between T2D and its intermediate phenotypic traits suggests their potential role in beta cell failure. These genes enriched the adipo-centric pathways and were expressed in both pancreatic and adipose tissue and, therefore, might be one of the potential targets for future antidiabetic treatment.

## 1. Introduction

The current epidemic of diabetes presents a challenge to the global healthcare system. It has been speculated that by 2045 over 629 million people will have diabetes [1], and as per the International Diabetes Federation, type 2 diabetes (T2D) would account for over 90% of these patients. This alarming situation can be attributed to our inadequate understanding of the diverse pathogenic processes involved at the molecular level that converge into two well-defined intermediate glycemic traits—impaired insulin secretion by the pancreatic beta cells and insulin resistance in peripheral tissues. Understanding these underlying mechanisms wouldhugely benefit in the designing of better antidiabetic agents to treat T2D and its associated micro-vascular (nephropathy, neuropathy, and retinopathy) and macro-vascular (atherosclerosis and cardiovascular) complications.

The etiology of T2D is complex; here, a large number of susceptibility genes in varying combinations conspire with environmental factors to give rise to the final pathophysiological pathways of insulin resistance and secretion defects. Even though several hundred susceptibility loci for T2D have been identified using genome-wide association studies (GWAS), they account for only about 20%–30% of the heritability of T2D, with the remaining “missing heritability” being attributed to rare or small effect variants, epistasis, gene–environment interaction epigenetic changes, etc. [2].

However, the utility of GWAS in the functional interpretation of T2D pathogenesis is still debatable because a fraction of the identified genetic loci is limited to the geographical and ethnic background of participants. Furthermore, a majority of them did not clearly converge on the functional categories consistent with known aspects of T2D pathophysiology, i.e., impaired insulin secretion and insulin resistance in peripheral tissues—the two hallmark glycemic intermediate traits of T2D.

These intermediate traits should be expected to show less locus heterogeneity and epistasis than the disease-associated loci themselves and investigations focusing on the functional aspects of these loci could shed some light on the underlying molecular pathophysiological processes of T2D in a better way. Additionally, to understand the genome-to-phenome mechanism in T2D, integrating GWAS results with genome-wide transcriptome profiles in the pancreas and other insulin-responsive tissues can also prove valuable, as the differentially expressed genes (DEGs) identified in these tissues between non-diabetics and diabetics could possibly lead to identification of the underlying fundamental biological process—perhaps beyond the genetic differences between individuals, thus circumventing the limitation of GWAS.

Previously we have used a systems biology approach to unravel genome-to-phenome correlation in T2D in Asian Indians [3]. We found that the physical and genetic interaction networks of the GWAS genes showed robust enrichment of insulin signaling and other T2D pathophysiology-related pathways, including insulin secretion. Additionally, we generated genome-wide expression profiles of adipose tissue from non-diabetic and diabetic patients. Remarkably, the differentially expressed genes showed a significant overlap with the network genes with the intersection showing enrichment of insulin signaling and other pathways consistent with T2D pathophysiology. Therefore, studying the overlap between networks of GWAS-identified genes and DEGs in disease-related tissues could shed some light on the underlying molecular mechanisms and genome-to-phenome correlation in diabetes.

Recently, to disseminate and analyze T2D-linked human genetic information and to accelerate the development of new therapies, a knowledge portal (http://www.type2diabetesgenetics.org/) has been established by National Institute of Health, USA. It lists a total of one hundred and thirty-two T2D-effectors genes, which were computationally predicted by synthesizing multiple genetic, regulatory, and perturbational evidences in support of their role in T2D pathophysiology (see Appendix A). Therefore, they can be considered as high confidence seed T2D genes. This portal also contains GWAS data on associations between genetic variants and glycemic traits, such as fasting glucose, Hb1Ac, Two Hour Glucose, Fasting Insulin, Insulin Sensitivity, HOMA-IR, HOMA-β, and OGTT. We selected HOMA-IR and HOMA-β [4] as measures of insulin resistance and insulin secretion, respectively, due to the availability of large sets of GWAS data on these traits, comprising over 46,186 non-diabetic participants [5].

Our proposed hypothesis is that the genes showing an association with these intermediate phenotypic traits should be involved in overall glucose homeostasis and if we locate these genes in a T2D-specific protein–protein interaction network they might show overlap with DEGs in diabetes-related tissues like pancreas, muscle, and adipose. We also hypothesize that a functional analysis of the subnetworks associated with these genes could help understand the molecular mechanisms underlying insulin resistance and unraveling genome-to-phenome correlation in T2D subjects. As it is ethically not possible to simultaneously do GWAS for T2D and its intermediate phenotypic traits and transcription profiling of related tissues in a single experiment, in silicoanalysis of the available information in databases could be an alternative for such studies; however, it demands experimental validation.

Therefore, in the present study, we attempted to further the insight into the molecular pathophysiology of T2D by (1) integrating “predicted T2D-effectors” with genes associated with HOMA-IR and HOMA-β using a network-based approach; (2) analyzing the HOMA-β and HOMA-IR subnetworks to identify the percentage of overlapping genes with DEGs in diabetes-related tissues to unravel the relative contribution of these organs in the overall T2D pathophysiology in terms of its intermediate phenotypic traits (HOMA-β/IR); and (3) functional analysis of these networks to identify pathogenic mechanisms leading to T2D.

## 2. Materials and Methods

We obtained one hundred and thirty-two predicted T2D-effector genes from the T2D knowledge portal. A T2D-intractome was then constructed in cytoscape [6] taking these effector genes as *seed nodes* along with their first-degree neighbors from a high confidence human protein–protein network obtained from ConsensusPathDB and StringDB using the PhenomeScapeapp [7].

Summary-level GWAS meta-analysis results for HOMA-β and HOMA-IR were also retrieved from the T2D Knowledge portal and *p*-values associated with each genetic variant were converted to gene-based empirical association *p*-values using Versatile Gene-based Association Study 2 (VEGAS2) [8], which is available online (https://vegas2.qimrberghofer.edu.au/).

To locate subnetworks or modules associated with insulin secretion and insulin signaling-related cellular processes, we mapped the HOMA-β and HOMA-IR-associated genes (*p* < 0.05) on the T2D-interactome and then used the Cytoscape app—jActiveModules [9]—to search for individual subnetworks for HOMA-β and HOMA-IR genes by taking gene-level *p*-values as numerical node attributes.

We also obtained a list of DEGs from the Type 2 Diabetes Mellitus Associated Complex Disorders (T2DiACoD) portal (http://t2diacod.igib.res.in/) [10] for pancreas and insulin-responsive tissues:adipose and muscle from the Normal Glucose Tolerant (NGT) vs. T2D microarray studies(see Appendix A). As each tissue had three microarray studies, composite DEGs were obtained by compiling DEGs obtained from each study. We then calculated the percentage of composite DEGs that were also associated with glycemic traits to look for the relative contribution of each tissue in diabetes pathophysiology. Further, the DEGs associated with the HOMA-β trait in each tissue were also analyzed for their functional enrichment of KEGG pathways using WebGestalt (WEB-based Gene SeTAnaLysis Toolkit) [11].

To assess the functionality of proteins in these subnetworks, we used theEnrichR method [12] for pathway enrichment analysis against the KEGG database [13] through PathwayConnector (http://bioinformatics.cing.ac.cy/PathwayConnector/) [14], which provides functionality to construct complementary pathway-to-pathway networks and subnetworks based on a reference KEGG pathway network. These pathway subnetworks were then explored in the light of existing literature to underpin the pathogenesis of type 2 diabetes.

## 3. Results

A T2D interactome of 2222 proteins was constructed for the T2D-effector genes (proteins). VEGAS2 provides 8135 genes associated with HOMA-βand 299 genes associated with HOMA-IR GWAS-variants, each with an empirical *p*-value < 0.05 (see Appendix A). This large difference in the number of genes between HOMA-β and HOMA-IR strengthens the hypothesis that defects in insulin secretion are caused by a large number of genes and may have a large genetic component, while insulin resistance may largely be induced by environmental factors [15].

A total of 1360 HOMA-β-associated genes overlapped with the T2D interactome showing that almost 61% of the network nodes are functionally associated with the insulin secretion-related trait of T2D, while only 24 HOMA-IR (∼8%) genes were mapped to the T2D interactome network. Out of these 24 HOMA-IR genes, 20 genes (*ACVRL1, AGTRAP, ATRNL1, CD44, CDKN1A, DAG1, GRB10, INADL, IRS1, MAD2L2, PLCG2, PTEN, RHOA, SIRT1, SNAP25, SNCA, TET2, TGM2, WWOX,* and *ZMIZ1*) were common between both the traits. In other words, barring the genes shared with HOMA-β, most of the HOMA-IR-associated genes showed no association with the T2DM trait.

Due to the central role of the pancreas in diabetes pathology, we compared the list of 295 DEGs obtained from the pancreatic transcriptomic analysis from T2DiACoD with that of adipose (253 DEGs) and skeletal tissues (275 DEGs). Adipose tissue shared 119 (53.45%) and skeletal tissue shared 147 DEGs (34.8%) with those in pancreas. A total of 62 DEGs were common across all the tissues.

We also calculated the percentage overlaps of these DEGs with the HOMA-IR- and HOMA-β-associated T2D-interacotme genes (Table 1) and found that approximately 20%–25% of these DEGs overlapped with the HOMA-β-associated network. A total of 16 common DEGs across all the three tissues were found to be associated with the HOMA-βtrait in our T2D-interactome (*CDKAL1, ACVRL1, APOE, SYVN1, SPP1, SCARB1, CNDP2, DNMT1, SRC, TRIB3, EGFR, MAPK14, ATP2A2, IGFBP5, D1,* and *PRKAA1).* Interestingly, 13 of these genes already have been reported in diabetes or its related phenotypes in the DisGeNET database [16] and therefore validated our network-based bioinformatics approach (Figure 1).

Due to their differential regulation in T2D-associated tissues along with their association with insulin secretion-related traits, our insilico analysis points towards a major role of these genes in diabetes pathophysiology, with high confidence.

We also did a functional enrichment analysis of DEGs in pancreatic, skeletal, and adipose tissue showing overlap with the HOMA-β-associated genes, and found that they enriched the *Insulin Signaling* and *Insulin Resistance pathways* in skeletal muscles and adipose tissues (see Appendix A). Adipose tissue also enriched the *Adipocytokine signaling pathway*, pointing towards the role of adipose tissue in insulin secretion defects of diabetes measured as HOMA-β. Besides, pancreatic DEGs also enriched *Cholesterol metabolism*, which further substantiates the adipocentric origin of the overall pathology.

Subnetwork analysis of genes showing association with HOMA-IR and HOMA-β in GWAS and overlap with the interactome of the T2D-associated genes was also done. The jActiveModules constructed five subnetworks for each of these traits. For functional analyses, a HOMA-β network of 291 nodes containing a maximum 245 mapped genes and a HOMA-IR network of 32 nodes with 18 mapped genes was selected for functional analysis.

EnrichR primarily enriched 144 KEGG pathways, including various diabetes-related pathways (*Insulin signaling pathway, Type II diabetes mellitus, AGE-RAGE signaling pathway in diabetic complications, Insulin resistance, Maturity onset diabetes of the young,* and *Insulin secretion*) with extremely significant *p*-values. The top 15 pathways, including two T2D-related pathways—*Insulin signaling pathway* (Rank 7) and *Type 2 diabetes mellitus* (Rank 12)—were selected for subsequent complementary pathway-to-pathway network analysis. This post-pathway analysis connected these pathways with 13 other complementary pathways and improved the rank of *Type2 diabetes mellitus* pathway from 12 to 2 (Table 2; Figure 2).

Some of these pathways have already been implicated in T2D pathophysiology viz. the *MAPK signaling pathway, PI3K-Akt signaling pathway, mTOR signaling pathway,* and *Adipocytokine signaling pathway*. Pathways related to cancer in various organs and bacterial/viral infections were also observed along with two new anti-cancer drug metabolic pathways—*EGFR tyrosine kinase inhibitor resistance* and *Platinum drug resistance*.

To understand the functional theme of T2D, PathwayConnector also provides a sub-grouping of complementary pathway-to-pathway networks using a community structure detection algorithm, which provided three well-defined clusters.

Three pathways—*Non-alcoholic fatty liver disease (NAFLD), Type 2 Diabetes Mellitus,* and *Adipocytokine signaling*, were clustered together, pointing towards the adipo-centric origin of both morbidities. The prevalence of NAFLD in patients with T2D is high [17]. Failure to store extra fat in the adipose tissue leads to its ectopic deposition in the liver with concurrent development of insulin resistance in peripheral tissues. A low circulating blood level of adiponectin—an adipocytokine, which has both hepatoprotective as well as anti-inflammatory activity—has been reported in NAFLD and T2D [18].

The second cluster constitutes various cell signaling pathways with reported involvement in T2D pathophysiology. Most of the metabolic effects of insulin signaling are mediated by the *PI3K-Akt signaling pathway*; its activation leads to the disposal of blood glucose into insulin-sensitive tissues (through promoting translocation of glucose transporter GLUT2/4), suppression of hepatic glucose production, andreduction in circulating free fatty acids (FFA), by promoting their deposition in adipose depots. Excess energy consumption has been reported to cause an imbalance in this pathway, leading to a T2D pathogenic cascade as follows: increased adipose tissue lipolysis, increase in FFA concentration, and impaired insulin signaling and consequent development of T2D [19]. This pathway is also one of the most frequently dysregulated signaling pathway in human cancers; this dysregulation has largely been attributed to the inactivation of the functionof a tumor suppressor gene Phosphatase and tensin homolog (PTEN), which is a negative regulator of the PI3K pathway and is associated with its increased activity resulting in various human cancers as also reflected by our complement pathway analysis.

Another enriched pathway *mTOR signaling pathway* is also connected with *PI3K-Akt signaling* and plays a key role in regulating glucose and lipid metabolism besides cell growth. The mTOR/receptor complex is activated by Akt and phosphorylase S6 Kinase, which has been reported to cause insulin resistance by serine phosphorylation of insulin receptor substrate-1 (IRS-1), eventually disrupting *PI3K-Akt signaling* [20].

The *Mitogen-activated protein kinases (MAPK) signaling pathway* is another major signaling pathway that affects insulin signaling via another enriched pathway the *Ras signaling pathway* and mediates the anabolic effects of insulin signaling, such as cell growth and differentiation. MAPKs, mainly extracellular signal-regulated kinase (ERK),are involved in the proliferation and differentiation of adipocytes. It has been reported that a high-fat diet induced hypertrophy in 3T-3L adipocyte cells, disturbing the normal physiological role of the *MAPK signaling pathway* and leading to enhanced lipolysis and insulin resistance in adipose tissue. Pharmacological inhibition of these kinases may provide a potential new strategy for the treatment of insulin resistance and type 2 diabetes [21].

We also investigated two new pathways—*EGFR tyrosine kinase inhibitor resistance and Platinum drug resistance*—in the third cluster amongst various cancer-related pathways.

Inflammation in adipose tissue, i.e., “adiposopathy”, has been regarded as the main pathogenic pathway that leads to insulin resistance in peripheral tissues and subsequent development of T2D; it is characterized by infiltration of macrophages in adipose depots and is attributed to epidermal growth factor receptor (EGFR)-mediated chemotaxis and proliferation of monocytes and macrophages. It has been reported that EGFR tyrosine kinase inhibitors improve glucose tolerance and insulin action in high-fatdiet-fed mice [22]; therefore, these agents can be used as an effective antidiabetic therapy.

Another drug pathway—*Platinum drug resistance*—further underlies the pharmacological potential of PPARγ agonists as antidiabetic agents as these drugs also exert their anticancer affect by acting as PPARγ agonists [23] similar to the thiazolindinedione classes of antidiabetics [24]. PPARγ is a transcription factor that functions as a master regulator of fat development in the body.

For the insulin resistance module of 32 genes, EnrichR enriched 136 KEGG pathways and further connected the top 15 pathways with 12 complementary pathways, including two new pathways—*Fluid shear stress and atherosclerosis* and *EGFR tyrosine kinase inhibitor resistance* (Table 3).

Inflammation related pathways—*Leukocyte trans-endothelial migration* and *Chemokine signaling pathway*—strongly suggest that these are indeed the primary events taking place in the adipose tissue and are involved in the genesis of type 2 diabetes.

Some metabolic pathways—*FoxO signaling pathway, mTOR signaling pathway, HIF-1 signaling pathway, PI3K-Akt signaling pathway, ErbB signaling pathway, MAPK signaling pathway,* and *p53 signaling pathway*—were also found to be enriched.

The *FoxO signaling pathway* is mediated by transcription factor FOXO that stimulates the transcription of the genes that inhibit cell proliferation or induce cell death. The *PI3K-Akt signaling pathway* promotes cell proliferation and survival by inactivating FOXO. The *HIF-1 signaling pathway* is mediated by hypoxia-inducible factor-a transcription factor, whose high level due to obesity-induced hypoxic condition in adipose tissue may give rise to inflammation in these depots [25]. The *p53 signaling pathway* is associated with DNA damage, which might be the result of oxidative stress in adipocytes due to increased lipolysis in diabetic conditions [26]. The *ErbB signaling pathway* has also been reported to cause diabetes-associated micro- and macro-vascular complications due to its increased activation [27].

## 4. Discussion

Understanding the molecular pathogenesis of complex diseases, such as type 2 diabetes, cardiovascular complications, Alzheimer’s, Parkinson’s, and various types of cancers, is challenging, hindering the development of an effective treatment. Assessment of disease-related intermediate phenotypic traits is therefore an important initial step towards any systematic genomic study [28]. In the present study, we hypothesized that genes significantly associated with the intermediate glycemic traits HOMA-IR and HOMA-β would likely help in identifying subnetworks of T2D protein–protein networks that may be targeted for understanding the pathogenic mechanisms leading to the disease, and also provide a clue for potential drug targets for pharmacological interventions. To the best of our knowledge, no published reports in the English literature have attempted studying the overlap between the networks associated with diabetes and its intermediate phenotypic traits. It is hypothesized that the molecular network shared by diabetes and its intermediate phenotypic traits is worthy to be called the most fundamental molecular network of diabetes. If the genes in this network are differentially expressed in any specific organs, this would further point towards the molecular pathology of diabetes. However, pathway-based analysis for deciphering the molecular mechanism of complex diseases is not a trivial task since the functional linkage of various human pathways is still unexplored, and it poses a problem in relating these pathways with disease phenotype. We therefore used a novel method that was implemented in PathwayConnector for the construction of pathway-to-pathway complementary networks. It facilitated us to interpret the functional theme in an unbiased and systematic way.

In this study we found a 61% overlap between the diabetes network and HOMA-β network, whereas this overlap was less than 2% for the HOMA-IR network. Despite this limitation, our findings are consistent with the already established fact that diabetes susceptibility genes play a major role in the impairment of pancreatic beta cell function, while insulin resistance is predominantly contributed by the environmental factors [15]. However, two additional findings of this study need to be highlighted. Firstly, on functional analysis, the HOMA-β network genes overlapping T2D interactome encompasses both the insulin secretion-related pathways as well as molecular pathways related to insulin signaling and fat metabolism, which also included some of the pathways otherwise responsible for adipose tissue dysfunction underlying diabetes and metabolic syndrome. Secondly, 80% of the genes in the HOMA-IR network overlapping with the T2D interactome were also overlapped with genes in the HOMA-β network. In other words, some of the genes associated with intermediate traits of insulin secretion (HOMA-β) and action (HOMA-IR) overlapped with each other. This association of the HOMA-β-associated gene network with insulin action and adipose tissue dysfunction was further supported by the finding that significant overlap between these genes and the differentially expressed genes in pancreas, adipose, and skeletal tissue in diabetics occurred.

Interestingly, there was also a significant overlap between the DEGs in these three tissues. Therefore, this bioinformatic analysis points towards a common underlying molecular mechanism for insulin resistance and secretion defects, which is shared by pancreatic, skeletal, and adipose tissues. One can assume these genetic mechanisms to be the “core molecular mechanisms” of pathogenesis of T2D and might be the most specific target for preventive and therapeutic strategies. These findings are consistent with those of our previous report where we examined the enrichment of pathways in genes identified in T2D GWAS. We found that some of these genes, specifically those at the lower significance threshold, showed enrichment of the insulin secretion-related pathway, but the physical and genetic interaction network of these genes showed robust enrichment of both insulin signaling and insulin secretion pathways.

The finding of this study, that a functional analysis of the subnetworks encompassing HOMA-β-associated GWAS signal-harboring genes enriched pathways of especially fat metabolism, with a central role of adipose tissues, support the “lipotoxicity” theory of beta cell failure in T2D, where heightened free fatty acid flux results from adipose tissue leads to increased insulin resistance. Indeed, several studies have demonstrated that obese type 2 diabetic patients presented a disturbed adipokine profile, which seems to be an important link between adiposity (i.e., insulin resistance), beta cell dysfunction, and T2D [29,30]. There is pharmacological evidence that drugs like insulin and thiazolidinediones significantly suppress the free fatty acid flux and improves beta cell functions in diabetes. Therefore, a genetically determined (as suggested by results of this bioinformatics analysis) adipo-centric pathway of pancreatic beta cell failure and insulin resistance could prove to be one of the most important targets for intervention for preventing and treating T2D. Moreover, identification of circulatory biomarkers of this common “genome-to-phenome pathway” underlying both beta cell failure and insulin resistance, could help in establishing “adiposopathy” as a clinical entity with the scope for intervention before beta cell dysfunction begins, as a pre-primary prevention of diabetes. The findings of this analysis have potential future application in the clinical practice of diabetes in the sense that diagnosing and treating adipocentric pathophysiological pathways found in this analysis could help in not only preventing diabetes and metabolic syndrome, but also in the preservation of pancreatic beta cell reserves. In clinical practice, preservation of beta cell function during most of the natural history of disease can be expected to be associated with better metabolic control and, therefore, a reduced burden of both microvascular and macrovascular complications of diabetes. However, a fundamental limitation of the present analysis is that it is based on literature-derived evidence and needs to be proven through actual laboratory experimentations. Furthermore, the GWAS data of glycemic traits we have used in this analysis tend to be ethnicity specific; however, we attempted to minimize this effect by integrating this data with relevant transcriptomic and protein–protein interaction data.

## 5. Conclusions

We have proposed a generic bioinformatics workflow that can be used with a high confidence computational gene list as well as a prioritized gene list, generated via high-throughput omics experiments. Our analysis endorses the adipocentric origin of type 2 diabetes, including beta cell dysfunction; appreciation of this fact will likely ideate the biomedical scientist to develop therapies with emphasis on improvement of fat biology in conjugation with maintenance of blood glucose levels. An adipocentric mechanism of T2D also underscores the significance of lifestyle improvements in curing metabolic disturbances like T2D, cardiovascular risk, etc.

## Figures and Tables

**Figure 1 biomolecules-10-00601-f001:**
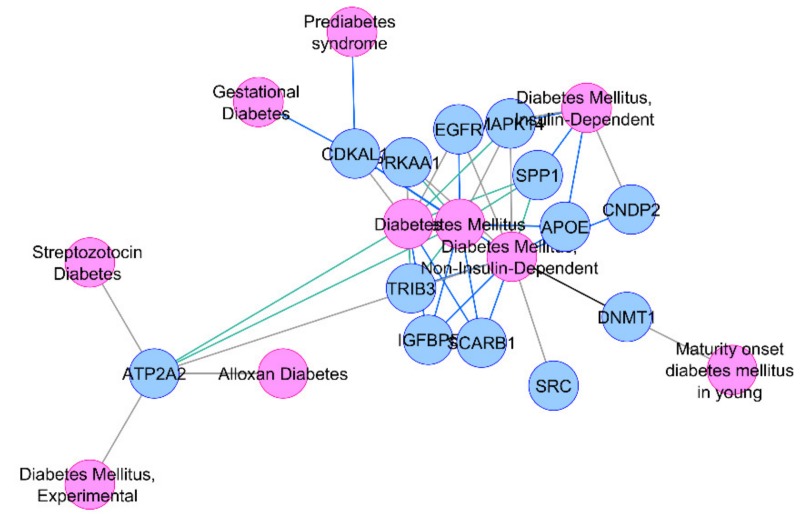
Association of thirteen common HOMA-β differentially expressed genes (DEGs) with diabetes or its related phenotypes.

**Figure 2 biomolecules-10-00601-f002:**
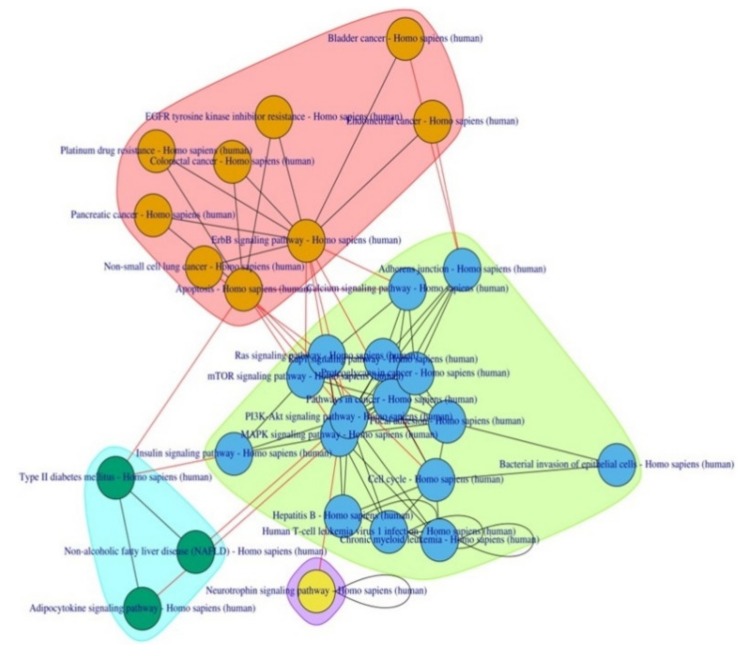
Pathway-to-pathway network enriched by the HOMA-β sub network.

**Table 1 biomolecules-10-00601-t001:** DEGs in various tissues and their overlap with HOMA-βand HOMA-IR GWAS genes.

S. No.	Tissue	# of DEGs Obtained (*p* < 0.05)	Overlap with HOMA-β	Overlap with HOMA-IR
1	Pancreas	295	74 (~25%)	7
2	Skeletal	275	63 (~22.7%)	4
3	Adipose	253	55 (~21.7%)	3

**Table 2 biomolecules-10-00601-t002:** PathwayConnector—complementary pathway networks enriched by the HOMA-β network.

S. No	Pathway	*p* Value	Common Pathways	Common Genes	Genes Found	Pathway Ratio	Rank
1	Adherens junction	4.61653 × 10^−26^	7/4246	19/72	19/291	1.649 × 10^−3^	3
2	Type II diabetes mellitus	1.22599 × 10^−13^	5/4246	13/46	13/291	1.178 × 10^−3^	12
3	Chronic myeloid leukemic	1.35055 × 10^−17^	8/4246	18/76	18/291	1.884 × 10^−3^	4
4	Pathways in cancer	4.61653 × 10^−26^	23/4246	52/526	52/291	5.417 × 10^−3^	1
5	ErbB signaling pathway	7.28619 × 10^−15^	10/4246	17/85	17/291	2.355 × 10^−3^	10
6	Bacterial invasion of epithelial cells	3.71078 × 10^−13^	7/4246	15/74	15/291	1.649 × 10^−3^	15
7	Proteoglycans in cancer	1.21687 × 10^−21^	14/4246	30/201	30/291	3.297 × 10^−3^	2
8	Neurotrophin signaling pathway	4.91599 × 10^−17^	6/4246	21/119	21/291	1.413 × 10^−3^	5
9	Pancreatic cancer	5.48038 × 10^−13^	10/4246	14/75	14/291	2.355 × 10^−3^	16
10	Colorectal cancer	2.16852 × 10^−13^	7/4246	19/72	19/291	1.649 × 10^−3^	3
11	Insulin signaling pathway	1.09957 × 10^−15^	5/4246	13/46	13/291	1.178 × 10^−3^	12
12	Focal adhesion	2.55996 × 10^−15^	8/4246	18/76	18/291	1.884 × 10^−3^	4
13	Hepatitis B	2.3797 × 10^−16^	23/4246	52/526	52/291	5.417 × 10^−3^	1
14	Human T-cell leukemic virus 1 infection	9.37 × 10^−15^	10/4246	17/85	17/291	2.355 × 10^−3^	10
15	Rap1 signaling pathway	6.90707 × 10^−15^	7/4246	15/74	15/291	1.649 × 10^−3^	15
16	EGFR tyrosine kinase inhibitor resistance	**-**	14/4246	30/201	30/291	3.297 × 10^−3^	**New**
17	Ras signaling pathway	3.086743 × 10^−13^	6/4246	21/119	21/291	1.413 × 10^−3^	5
18	Non-small cell lung cancer	1.41 × 10^−6^	10/4246	14/75	14/291	2.355 × 10^−3^	16
19	Endometrial cancer	7.88 × 10^−7^	10/4246	15/86	15/291	2.355 × 10^−3^	13
20	MAPK signaling pathway	1.13 × 10^−9^	7/4246	21/137	21/291	1.649 × 10^−3^	7
21	Adipocytokine signaling pathway	7.72 × 10^−7^	9/4246	24/199	24/291	2.120 × 10^−3^	8
22	PI3K-Akt signaling pathway	6.34 × 10^−9^	11/4246	21/163	21/291	2.591 × 10^−3^	6
23	Apoptosis	1.38 × 10^−7^	12/4246	25/219	25/291	2.826 × 10^−3^	11
24	Cell cycle	2.02 × 10^−6^	9/4246	24/206	24/291	2.120 × 10^−3^	9
25	Platinum drug resistance	**-**	10/4246	12/79	12/291	2.355 × 10^−3^	**New**
26	Non-alcoholic fatty liver disease (NAFLD)	1.36 × 10^−5^	9/4246	24/232	24/291	2.120 × 10^−3^	14
27	Bladder cancer	2.91 × 10^−3^	8/4246	10/66	10/291	1.884 × 10^−3^	69
28	mTOR signaling pathway	2.33 × 10^−4^	9/4246	8/58	8/291	2.120 × 10^−3^	64

**Table 3 biomolecules-10-00601-t003:** PathwayConnector—complementary pathway networks by HOMA-IR interactome.

S. No	Pathway	*p* Value	Common Pathways	Common Genes	Genes Found	Pathway Ratio	Rank
1	Proteoglycans in cancer	4.932838 × 10^−13^	14/4246	10/201	10/32	3.297 × 10^−3^	1
2	Glioma	8.665005 × 10^−12^	8/4246	7/75	7/32	1.884 × 10^−3^	2
3	Pathways in cancer	1.556996 × 10^−7^	23/4246	9/526	9/32	5.417 × 10^−3^	10
4	MicroRNAs in cancer	1.661089 × 10^−8^	7/4246	8/299	8/32	1.649 × 10^−3^	5
5	FoxO signaling pathway	1.437119 × 10^− 9^	13/4246	7/132	7/32	3.062 × 10^−3^	3
6	mTOR signaling pathway	2.470845 × 10^−6^	8/4246	7/153	7/32	1.884 × 10^−3^	18
7	HIF-1 signaling pathway	1.30885 × 10^−8^	9/4246	6/100	6/32	2.120 × 10^−3^	4
8	PI3K-Akt signaling pathway	9.186771 × 10^−7^	17/4246	7/354	7/32	4.004 × 10^−3^	14
9	Neurotrophin signaling pathway	3.280781 × 10^−8^	6/4246	6/119	6/32	1.413 × 10^−3^	6
10	Longevity regulating pathway	3.751219 × 10^−7^	8/4246	5/62	5/32	1.884 × 10^−3^	7
11	Melanoma	9.131152 × 10^−8^	7/4246	5/72	5/32	1.649 × 10^−13^	8
12	Bacterial invasion of epithelial cells	1.468947 × 10^−7^	7/4246	5/74	5/32	1.649 × 10^−3^	9
13	EGFR tyrosine kinase inhibitor resistance	-	10/4246	5/79	5/32	2.355 × 10^−3^	new
14	Focal adhesion	7.174111 × 10^−7^	9/4246	6/199	6/32	2.120 × 10^−3^	13
15	Longevity regulating pathway	5.389533 × 10^−8^	9/4246	5/89	5/32	2.120 × 10^−3^	12
16	Prostate cancer	2.852625 × 10^−7^	9/4246	5/97	5/32	2.120 × 10^−3^	11
17	Leukocyte transendothelial migration	1.163798 × 10^−6^	3/4246	5/112	5/32	7.065 × 10^−4^	15
18	ErbB signaling pathway	1.1 × 10^−5^	10/4246	4/85	4/32	2.355 × 10^−3^	20
19	Small cell lung cancer	0.000348	7/4246	4/93	4/32	1.649 × 10^−3^	37
20	Non-small cell lung cancer	0.003619	8/4246	3/66	3/32	1.884 × 10^−3^	57
21	MAPK signaling pathway	0.062537	7/4246	4/295	4/32	1.649 × 10^−3^	96
22	p53 signaling pathway	0.000181	4/4246	3/72	3/32	9.421 × 10^−4^	31
23	Colorectal cancer	0.004417	10/4246	3/86	3/32	2.355 × 10^−3^	61
24	Fluid shear stress and atherosclerosis		9/4246	3/139	3/32	2.120 × 10^−3^	new
25	Cell cycle	0.016751	4/4246	2/124	2/32	9.421 × 10^−4^	81
26	Apoptosis	0.021037	9/4246	2/136	2/32	2.120 × 10^−3^	87
27	Chemokine signaling pathway	0.035898	7/4246	2/190	2/32	1.649 × 10^−3^	94

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
