# Peer review of "Functional Interactomes of Genes Showing Association with Type-2 Diabetes and Its Intermediate Phenotypic Traits Point towards Adipo-Centric Mechanisms in Its Pathophysiology"

_biomolecules, 2020, doi:10.3390/biom10040601_

Round 1

Reviewer 1 Report

Please provide P-values for enrichment analyses in Tables 1 and 2. Without any statistical significance, we cannot judge whether the results provided are more probable than random matching.  After providing these measures of statistical significance, we can evaluate the contents more properly. 

Author Response

Dear Sir/ M'am,

Thank you very much for highlighting this important error. We have now mentioned the p-value in both tables 1-2. All results are based upon statistically significant p-value ≤ 0.05.

Sincerely Yours

Reviewer 2 Report

Dr. Aditya Saxena et al. used a network-based approach to assess the cellular processes associated with the protein-protein interaction sub networks in T2DM. They analyzed the overlap differentially expressed genes (DEGs) in pancreatic beta cells, muscle, and adipose tissue of diabetic patients. Functional gene interaction was elucidated by integrated pathways-analysis. The authors found many DEGs in all the selected tissues showing overlap mainly with HOMA-β sub network suggesting a role of these tissues in β-cell failure. 61% DEGs overlap between diabetes network and HOMA-β network, whereas only 2% of genes overlap with HOMA-IR network. The bioinformatic analysis revealed a common underlying molecular mechanism for insulin resistance and secretion defect shared by pancreatic beta cells, skeletal and adipose tissues. This study is somehow related to recently published work (a network metanalysis study) by the same group, see Current Genomics, 2018 Nov;19(7):630-666. Although, this literature-derived study has some limitation and needs to be experimentally proved (as stated by the authors), the authors provide comprehensive generic bioinformatic workflow as a foundation for more focused mechanistic research for beta cell dysfunction in diabetes that could help in discovery of novel therapeutic approaches to alleviate diabetes development and progression.

Author Response

Dear Sir/ M’am,

We are grateful to you for endorsing our study. Your comment encourages us to continue our research towards adipocentric basis of type 2 diabetes. Necessary improvements have also been made in language front.

Thanks.

Reviewer 3 Report

This study used a network-based approach to assess the cellular processes associated with the protein-protein interaction sub networks of glycemic traits - HOMA-β, and HOMA-IR. Their sub networks were further analyzed in terms of their overlap with the differentially expressed genes (DEGs) in pancreatic beta cells, muscle, and adipose tissue in diabetics. Authors found many DEGs in all these tissues showing overlap with HOMA-β sub network, suggesting a role of these tissues in β-cell failure. For understanding the functional theme of these sub networks, a pathway-to-pathway complementary network analysis was done, which identified various adipose biology-related pathways, containing genes involved in both insulin secretion and action.

Subject definitions and methods are optimal. Tables, Figure and references are adequate.

I have found this paper relevant to the field of this journal. I have only one minor comment.

Minor point:

  • In the discussion, line 263-264, you mention: “In this study we found a 61% overlap between diabetes network and HOMA-β network, 263 whereas this overlap was less than 2% for HOMA-IR network.” Can you exclude ethnic effect of this result?
  • And you continue, line 264-267: “Our findings are consistent with the already established fact that diabetes susceptibility genes play a major role in the impairment of pancreatic beta cell function, while insulin resistance is predominantly contributed by the environmental factors.” Can you add some citations?
  • The title is missing for the Supplementary Table S4.

I recommend this paper for acceptation after minor revision in the journal.

Author Response

Dear Sir/ M’am,

Thank you very much for conducting review of our MS and providing your valuable feedback for its improvement. I am herewith addressing your concerns for your kinds pursue:

  1. The limitation of GWAS data is that it is ethnicity specific however HOMA IR/ HOMA B network that we have derived for functional analysis also include genes T2D effectors genes, and genes that directly interact with them. Integration of these data with GWAS data would likely minimize ethnic effect.
  2. Proper citation has been added.
  3. The title in the supplementary table S4 has been edited.

Sincerely yours.

Reviewer 4 Report

This manuscript by Dr. Aditya Saxena showed the

Functional interactomes of genes showing association 2 with type-2 diabetes and its intermediate phenotypic 3 traits. Although the results are pretty interesting, it need mechanistic study of a specific gene.

Author Response

Dear Sir/ M’am,

Thank you very much for conducting review of our MS and providing your valuable feedback for its improvement. I do agree that genomic medicine based study should go down to the level of gene for providing mechanistic insight. However the primary objective of our study was to explore system level analysis of T2D-associated networks. I have added a figure displaying association of 13 genes (out of 16) in HOMA B network with T2D. As, these genes were identified by our network analysis and hence signify the approach used.

Sincerely yours.

Reviewer 5 Report

Overall comments:

This manuscript was used a network-based approach to assess the cellular processes associated with the protein-protein interaction sub networks of glycemic traits. There are several areas where the manuscript needs to be strengthened.

Specific comments:

1.Statement of Ethics: In addition to Institutional Review Board approval, authors should also state that subjects have given their written informed consent.

2.What is the originality and strengths of this case report? How physicians or policy makers could deliberate with patients or people based on the key findings of this manuscript?

3.What is the reliability and validity for the data analysis.

4.More discussion regarding the clinical practice of their findings would be important for the specific journal.

5.The authors should add the comments related to selection bias in this study to the perceived limitation subsection.

6.Some references should be updated.

Totally, I would like to congratulate the authors for the enthusiasm invested in this study. However, the manuscript does not reach the level of quality required for publication as original research without major revision in Biomolecules.

Author Response

Dear Sir/ M’am,

Thank you very much for reviewing our MS and providing your valuable feedback for its improvement.                                                                       

I am herewith addressing your concerns for your kinds pursue:

  1. Statement of ethics does not hold with this study as it is solely based on computational analysis of existing data i.e. GWAS, gene expression, protein-protein interactions, etc.
  1. & 4. In the discussion section, we have added a new para, highlighting the key findings of the MS-                                                                            The findings of this analysis have potential future application in clinical practice of diabetes in the sense that diagnosing and treating adipocentric pathophysiological pathways found in this analysis could help in not only preventing diabetes and metabolic syndrome, but also preservation of pancreatic beta cell reserves. In clinical practice, preservation of beta cell function during the most part of natural history of disease could be expected to be associated with better metabolic control, therefore, reduced burden of both microvascular and macrovascular complications of diabetes.                                                                                                               
  2. We have added a figure displaying association of 13 genes (out of 16 common DEGs across insulin associated genes) in HOMA B network with T2D and its related phenotypes. As, these genes have been identified by our network analysis and therefore signify the approach used.                    Furthermore, functional analysis of HOMA-B/ HOMA IR enriched cellular pathways, those have clearly been implicated to the known aspect of T2D pathophysiology. These phenotypes-suggestive pathways henceforth endorse the reliability and validity of our data analysis.
  1. We have added a new comment in the limitation subsection-      Furthermore, GWAS data of glycemic traits, we have used in this analysis tend to be ethnicity specific however we attempted to minimize this effect by integrating this data with relevant transcriptomic, and protein-protein interaction data.
  1. New references have been provided proper citation.

Sincerely yours

Round 2

Reviewer 1 Report

Please provide exact P-values, not  in the form of  P<0.05.

Author Response

Dear Sir/Ma'm,

we have now included P value of each pathway in both the tables in Manuscirpt. I am also enclosing the Enrichment results and pathway connector statistics of both HOMA-B, and HOMA-IR interactome for further varification. 

Thanks 

Reviewer 4 Report

.

Author Response

Dear Sir/M'am,

Thanks for reviewing our MS second time, we have further made an updation by including P values of enrichment results in the Table 2, and Table 3.

Thnaks again 

Reviewer 5 Report

The reviewer's comments have been adequately addressed. I have no further questions.

Author Response

Dear Sir/M'am,

I am greatful to you for approving our study.

Thanks 

This manuscript is a resubmission of an earlier submission. The following is a list of the peer review reports and author responses from that submission.

Round 1

Reviewer 1 Report

The manuscript needs to be thoroughly read for grammar and typographical errors and also for consistency in term usage. For example, either use sub-network or subnetwork, the same for HOMA-R and HOMA-IR.

Line 18 – Suggesting the role of what tissues? The only tissue mentioned is adipose tissue.

Line 30 – Reference needed when state a fact such as this

Line 47 – if these T2D effector genes are publicly available on the T2D Knowledge Portal then you need to delete ‘(unpublished)’ as this gives cause to how you came into possession of these data. Especially with the information that you are referencing is stated on the website.

Line 155 – what co-morbidities? NAFLD and T2D? This sentence is not clear.

Line 244 – need to start a second sentence at ‘Secondly…’

Line 252 – again which tissues? Are you calling beta cells a tissue or when you say tissues are you meaning the pancreas and adipose? This needs to be better clarified throughout the manuscript.

Reviewer 2 Report

Comments:

The manuscript topic is interesting-

Diabetes has become a global problem all around the world, and starts already in children population, therefore the proper and quick intervention is very important.

The aim of this study was to attempted to further the insight into T2D-pathophysiology by integrating “predicted T2D-effetors” with genes associated with HOMA-IR, and HOMA-b using a network-based approach. They analysed T2D-interactome, HOMA-b, and HOMA-IR subnetworks to identify the percentage of overlapping genes with 1). Differentially expressed genes (DEGs) in pancreas and adipose tissues in normal glucose tolerant (NGT) and type 2 diabetics (T2D) with Gene Expression Omnibus (GEO) Database and 2). The genes showing an elevated expression in these tissues as  reported in Human Protein Atlas. They attempted to unravel the relative contribution of these organs in the overall T2D pathophysiology in terms of its intermediate phenotypic traits (HOMA-b/IR) in this analysis.

Comments:

He T2D pathophysiologyis were well known.

The major concerns are about methodology and results. The methodology how the date were analyzed are completely missing, authors used different databases, but how they put together the data there is even single word.

There is a big ethical concern.

Second, how can by data from different populations putting together, another population data were from GWAS, another data were from gene expression in adipose tissue, proteome, metabolome studies?

Thereafter we cannot absolutely believe the results of such a mix.

Third, the formula IR HOMA is not a proper marker of insulin secretion ans insulin sensitivita. For the first one IVGTT and the second one hyperinsulinemic euglycemic clamp are gold standard. IR HOMA indices were very inaccurate.

The Ethics it is even not mentioned.

As authors of the manuscript did not write the main limitation of the study, therefore it is questionable if it is not enough to get appropriate statistical power.

Authors also want to find the potential new pathways to develop new anti-diabetic agents.

They propose to use the platinum-based drugs as anti-diabetic agents! Platinum-based drug are used as chemotherapy in oncology and have so many adverse effects, that it is not probable that this group of drugs can be used for a such a purpose!

The major concerns are the method and limitation of the study.

Therefore the results and conclusions are useless.

Overall, I rate this paper very low, and this type of manuscript should not be published nowhere.

Reviewer 3 Report

Since there are no clear explanations by which readers can reproduce the results, I cannot judge if the results are correct or not at all. Although there are long lists of tools and databases used, no clear information about how the authors made use of these tools at all.

Please explain, step by step, what the inputs and the outputs are to individual tools. Then I might be able to evaluate if the results are reasonable.